# Comorbidity and health-related quality of life in people with a chronic medical condition in randomised clinical trials: An individual participant data meta-analysis

Elaine W. Butterly[1], Peter Hanlon[1], Anoop S. V. Shah[2], Laurie J. Hannigan[3,4,5], Emma McIntosh[1], Jim Lewsey[1], Sarah H. Wild[6], Bruce Guthrie[6], Frances S. Mair[1], David M. Kent[7], Sofia Dias[8], Nicky J. Welton[4], David A. McAllister[1]*

1 School of Health and Wellbeing, College of Medical, Veterinary and Life Sciences, University of Glasgow, Glasgow, United Kingdom, 2 London School of Hygiene and Tropical Medicine, London, United Kingdom, 3 Nic Waals Institute, Lovisenberg Diaconal Hospital, Oslo, Norway, 4 Population Health Sciences, Bristol Medical School, University of Bristol, Bristol, United Kingdom, 5 Department of Mental Disorders, Norwegian Institute of Public Health, Oslo, Norway, 6 Usher Institute, University of Edinburgh, Edinburgh, United Kingdom, 7 Predictive Analytics and Comparative Effectiveness Center, Tufts Medical Center/Tufts University School of Medicine, Boston, Massachusetts, United States of America, 8 Centre for Reviews and Dissemination, University of York, York, United Kingdom

* david.mcallister@glasgow.ac.uk

**Data Availability Statement:** The data that support the findings of this study are available from Clinical Study Data Request and the Yale Open Data Access repositories but restrictions apply to the availability of these data, which were used under license for the current study, and so are not

## Abstract

### Background

Health-related quality of life metrics evaluate treatments in ways that matter to patients, so are often included in randomised clinical trials (hereafter trials). Multimorbidity, where individuals have 2 or more conditions, is negatively associated with quality of life. However, whether multimorbidity predicts change over time or modifies treatment effects for quality of life is unknown. Therefore, clinicians and guideline developers are uncertain about the applicability of trial findings to people with multimorbidity. We examined whether comorbidity count (higher counts indicating greater multimorbidity) (i) is associated with quality of life at baseline; (ii) predicts change in quality of life over time; and/or (iii) modifies treatment effects on quality of life.

### Methods and findings

Included trials were registered on the United States trials registry for selected index medical conditions and drug classes, phase 2/3, 3 or 4, had ≥300 participants, a nonrestrictive upper age limit, and were available on 1 of 2 trial repositories on 21 November 2016 and 18 May 2018, respectively. Of 124 meeting these criteria, 56 trials (33,421 participants, 16 index conditions, and 23 drug classes) collected a generic quality of life outcome measure (35 EuroQol-5 dimension (EQ-5D), 31 36-item short form survey (SF-36) with 10 collecting both). Blinding and completeness of follow up were examined for each trial.

Using trials where individual participant data (IPD) was available from 2 repositories, a comorbidity count was calculated from medical history and/or prescriptions data. Linear

publicly available. Data are however available by directly applying to these repositories via application process to their respective independent data access committees. All data released from the respective safe havens (Clinical Study Data Request and the Yale Open Data Access) has been made available at https://github.com/ChronicDiseaseEpi/como_qol_public.

**Funding:** DM is funded via an Intermediate Clinical Fellowship and Beit Fellowship from the Wellcome Trust, who also supported other costs related to this project such as data access costs and database licenses ("Treatment effectiveness in multimorbidity: Combining efficacy estimates from clinical trials with the natural history obtained from large routine healthcare databases to determine net overall treatment Benefits." - 201492/Z/16/Z. SD is supported by the Medical Research Council [grant no. MR/R025223/1]. LH is supported by the South-Eastern Norway Regional Health Authority (#2020023, #2019097). PH is funded through a Clinical Research Training Fellowship from the Medical Research Council (Grant reference: MR/S021949/1). The funders had no role in study design, data collection and analysis, decision to publish, or preparation of the manuscript.

**Competing interests:** I have read the journal's policy and the authors of this manuscript have the following competing interests. NJW has received honoraria for training and masterclasses from: Association of British Pharmaceutical Industries, Campbell Ireland, Centre for Global Development, NICE International and NICE Scientific Advice, All Wales Therapeutics and Toxicology Centre, University of Leuven. NJW has delivered training for Takeda, ICON plc, and University of Galway for which payment was made to her institution. DM has received funding for this work from the Wellcome Trust.

**Abbreviations:** ATC, anatomic therapeutic chemical; CSDR, Clinical Study Data Request; EQ-5D, EuroQol-5 dimension; IPD, individual participant data; MCS, mental component score; NCT, national clinical trial; PCS, physical component score; SD, standard deviation; SLE, systemic lupus erythematosus; TNF-α, tumour necrosis factor α; YODA, Yale University Open Data Access.

regressions were fitted for the association between comorbidity count and (i) quality of life at baseline; (ii) change in quality of life during trial follow up; and (iii) treatment effects on quality of life. These results were then combined in Bayesian linear models. Posterior samples were summarised via the mean, 2.5th and 97.5th percentiles as credible intervals (95% CI) and via the proportion with values less than 0 as the probability ($P_{Bayes}$) of a negative association.

All results are in standardised units (obtained by dividing the EQ-5D/SF-36 estimates by published population standard deviations). Per additional comorbidity, adjusting for age and sex, across all index conditions and treatment comparisons, comorbidity count was associated with lower quality of life at baseline and with a decline in quality of life over time (EQ-5D −0.02 [95% CI −0.03 to −0.01], $P_{Bayes}$ > 0.999). Associations were similar, but with wider 95% CIs crossing the null for SF-36-PCS and SF-36-MCS (−0.05 [−0.10 to 0.01], $P_{Bayes}$ = 0.956 and −0.05 [−0.10 to 0.01], $P_{Bayes}$ = 0.966, respectively). Importantly, there was no evidence of any interaction between comorbidity count and treatment efficacy for either EQ-5D or SF-36 (EQ-5D −0.0035 [95% CI −0.0153 to −0.0065], $P_{Bayes}$ = 0.746; SF-36-MCS (−0.0111 [95% CI −0.0647 to 0.0416], $P_{Bayes}$ = 0.70 and SF-36-PCS −0.0092 [95% CI −0.0758 to 0.0476], $P_{Bayes}$ = 0.631.

## Conclusions

Treatment effects on quality of life did not differ by multimorbidity (measured via a comorbidity count) at baseline—for the medical conditions studied, types and severity of comorbidities and level of quality of life at baseline, suggesting that evidence from clinical trials is likely to be applicable to settings with (at least modestly) higher levels of comorbidity.

## Trial registration

A prespecified protocol was registered on PROSPERO (CRD42018048202).

## Author summary

### Why was this study done?

➢ The EuroQol-5 dimension (EQ-5D) and SF-36 are questionnaire-based tools that combine measures of **physical and mental health** into overall quality of life scores. These scores are used in randomised controlled trials of drug treatments to estimate how treatments affect quality of life. These estimates then inform decisions about which treatments should be offered to people with specific conditions.

➢ **Multimorbidity, the presence of 2 or more conditions, makes diagnosis and treatment more complex and is associated with worse quality of life in some settings.**

➢ People with multimorbidity are underrepresented in clinical trials, and little is known about whether and how multimorbidity changes quality of life in clinical trials. This makes it difficult for clinicians and clinical guideline developers to determine how results from clinical trials should be applied to people with multimorbidity.

**What did the researchers do and find?**

➣ To address this uncertainty, we re-analysed data from existing clinical trials. Among 33,421 participants in 56 trials of new treatments for 16 different medical conditions, we used data on medication usage and medical histories to produce a comorbidity count (higher counts indicating more multimorbidity) for each individual.

➣ We then used statistical models to examine associations for comorbidity counts finding that higher comorbidity counts were associated with worse quality of life at trial entry and predicted a more rapid decline in quality of life over the course of the trial. However, having a higher comorbidity count did not change the effect of treatment on quality of life.

**What do these findings mean?**

➣ These findings suggest that where treatments improve quality of life for participants overall, they are similarly likely to do so for people with multimorbidity.

➣ Whether this is true for individuals with higher numbers of comorbidities or with conditions and treatments not included in our analyses remains uncertain.

➣ Nonetheless, our findings help inform clinical decision-making by reassuring clinicians, health economists, and guideline developers that **overall trial results can be used when considering how best to manage many people with multimorbidity.**

## Introduction

Measures of health-related quality of life evaluate the impact of treatments in ways that matter to patients [1], and, by informing economic models, are important underpinnings of clinical guideline recommendations and therefore clinical practice. Generic quality of life measures, such as the EuroQol-5 dimension (EQ-5D) [2] and the 36-item short form survey (SF-36) [3] are valid across all conditions. This means they can be used to prioritise treatments even across different conditions. Consequently, quality of life measures are often included in randomised clinical trials (hereafter trials).

Multimorbidity, where individuals have 2 or more health conditions, is common and important as it is strongly associated with higher mortality and hospitalisation rates [4,5], and treatments that have been shown in clinical trials to improve quality of life are prescribed to people with multimorbidity less frequently than those without any comorbidities [6,7]. Cross-sectional associations between multimorbidity and quality of life have been reported [8–22]. Health-related quality of life has been identified as an essential core outcome in multimorbidity research [23]. Despite this, few studies have examined whether multimorbidity predicts change in quality of life over time [24–29], and whether the effect of treatments on quality of life differ according to the presence and extent of multimorbidity is unknown. Consequently, clinicians, regulatory agencies, and guideline developers are uncertain as to the applicability of trial quality of life findings for people with multimorbidity.

In previous individual participant data (IPD) analyses from a set of 124 clinical trials ranging across a number of index conditions and treatment comparisons, we have previously shown that multimorbidity, although underrepresented is present among trial participants and

predicts increased rates of both serious adverse events and trial attrition [5,30,31]. Using the 56 trials from this set for which measures of quality of life are available, we now aim to determine whether comorbidity count, which is increased in multimorbidity, predicts change in quality life, and whether treatment effects on quality of life differ by comorbidity count at baseline.

## Methods

This study is reported as per the Preferred Reporting Items for Systematic Reviews and Meta-Analyses (PRISMA) guideline (S1 PRISMA Checklist).

### Design

We performed a meta-analysis of trial IPD to determine the associations between comorbidity count and (i) quality of life at baseline; (ii) change in quality of life at trial follow-up; and (iii) the effect of treatment on change in quality of life at trial follow-up. Each analysis was done in 2 stages to account for the fact that trial data were stored securely on separate platforms and could not be analysed together in a single model. In the first stage, associations for individual trials were modelled within each secure trial analysis platform. In the second stage, the coefficients (and standard errors) from the first stage were meta-analysed using Bayesian linear models. We used a method that allowed partial pooling across index conditions and drug treatment comparisons in order to obtain overall drug treatment comparison specific and index-condition specific estimates of these associations.

### Trial inclusion

Industry sponsored randomised trials with available IPD were identified using a prespecified protocol (Prospero CRD42018048202) [32]. The full selection process and analysis plan has been documented previously [30].

In brief, the United States clinical trial registry at clinicaltrials.gov [33] was searched for eligible studies meeting the following criteria; randomised phase 2/3, 3 or 4 trials studying preselected drug classes used to treat or prevent 23 selected long-term medical index conditions, registered from January 1990 until the dates of initial IPD access on 21 November 2016 and 18 May 2018, for Clinical Study Data Request (CSDR) and Yale University Open Data Access (YODA) repositories, respectively. Trials were excluded where they examined treatment for neoplastic, infectious, affective, psychotic, or developmental disorders. Trials were included if they were registered to clinicaltrials.gov [33] and had IPD availability within CSDR or YODA IPD repositories at the time of the original data request to these platforms. To make efficient use of analyst time (both ours and that of the trial sponsors who anonymised the IPD), included trials were limited to those with ≥300 participants. Given our focus on comorbidity, trials were excluded if they limited participants to those aged less than 60 years old. For the current analysis, this set of trials was restricted to those with at least 1 generic health-related quality of life measure. As our aim in the current analysis was to examine the impact across a range of index conditions, we did not include trials which only included condition-specific measures of health-related quality of life (e.g., St. George's Respiratory Questionnaire). Included trials were categorised based on index medical condition and the World Health Organization Anatomic Therapeutic Chemical (ATC) drug class of the intervention drug [34].

### Comorbidity data

A comorbidity count was calculated for each trial participant. Twenty-one conditions were included in the count: cardiovascular disease, chronic pain, arthritis, affective disorders, acid-

related disorders, asthma/chronic obstructive pulmonary disease, diabetes mellitus, osteoporosis, thyroid disease, thromboembolic disease, inflammatory conditions, benign prostatic hyperplasia, gout, glaucoma, urinary incontinence, erectile dysfunction, psychotic disorders, epilepsy, migraine, parkinsonism, and dementia.

The methods used to derive this count have been described previously [30,35]. Briefly, where only data on concomitant medication was available, specific ATC codes were used to define conditions (e.g., use inhaled of corticosteroids indicated the presence of asthma or chronic obstructive pulmonary disease). Where medical history data was also available, specific medical dictionary for regulatory activities (MedDRA) codes were additionally used to identify individuals with any of the 21 conditions. The comorbidity counts were summed for each participant. The full list of ATC and MedDRA codes and the analysis code used to derive the comorbidity count are available at the project's GitHub repository (https://github.com/ChronicDiseaseEpi/como_qol_public).

### Outcome measures

Trials reported EQ-5D, SF-36, or both. EQ-5D is a preference-based generic health-related quality of life questionnaire with 5 domains (mobility, self-care, usual activities, pain/discomfort, and anxiety/depression) and a global measure of current health (measured via a visual analogue scale). Given EQ-5D is preference based, incorporating values for health outcomes, it can be used in cost-effectiveness analyses [36]. The domains are scored on either a 3 or 5 level scale (EQ-5D-3L or EQ-5D-5L). Since trials in our analysis included either type, but not both, we mapped each to a single EQ-5D index value (using United Kingdom population-based value sets for EQ-5D-3L and EQ-5D-5L, respectively, to obtain a single value that is comparable across all the EQ-5D trials [37]). EQ-5D index can range from negative values (reflecting a state felt to be "less preferable than death"), through 0 being "a state as bad as death" to 1 being "perfect health" [2].

The SF-36 questionnaire is also a commonly used generic quality of life instrument and includes 36 questions over 8 domains. Each domain score provides a weighted summary value of the questions within that domain. These domain scores are then summarised using a physical component score (PCS) and a mental component score (MCS). As is standard, we calculated these by first standardising the scales to z scores, then aggregating the physical and mental domains and transforming these into summary t scores ranging from 0 to 100 with higher scores denoting higher quality of life [3]. SF-36-PCS and SF-36-MCS scores are known to be correlated and therefore findings for each should not interpreted as if they provided independent verification [38].

### Treatment arms

Treatment arm comparisons were specified prior to undertaking the quality of life analyses. For multi-arm trials, the most extreme arms were selected for comparison (e.g., if different dosages were used, the highest dose was compared to placebo or usual care—e.g., canagliflozin 300 mg, rather than 100 mg, versus placebo). Where placebo or usual care was included as a treatment arm, this was selected as the comparator. Otherwise, we chose the arm with the oldest treatment as the comparator arm.

### Statistical analysis

Summary statistics were calculated for each index condition for the available EQ-5D and SF-36 trials including age (mean and standard deviation; SD), sex (number and %), and comorbidity count (mean and SD) and proportion with 0, 1 or 2, or more comorbidities.

Full descriptions of the modelling are provided in the supplementary (S1 Modelling description) and are described briefly below.

(i) *Association between comorbidity count and quality of life at baseline*

In linear regression models, for each trial and each measure, baseline quality of life was modelled adjusting for age (per 15-year increment, which was close to the SD for most trials), sex (male versus referent group of females), and comorbidity count (per additional comorbidity). The effect measure estimates and associated standard errors for each model were then exported from the YODA and CSDR secure analysis platforms. In order to convert the measures onto a similar scale, we standardised each; we did so by dividing the estimates and standard errors by published estimates of the standard deviation (EQ-5D-index– 0.23 [39], SF-36-PCS—9.08, and SF-36-MCS– 10.16 [40]).

The effect measure estimates for comorbidity count (adjusted for age and sex) terms were then separately meta-analysed in Bayesian linear regression models. We used Bayesian models since these allowed partial pooling across index conditions and treatment comparisons and because they allowed us to obtain credible intervals for estimates at the level of index condition and treatment comparison directly from the posterior without a need for post hoc calculations. We fitted a range of meta-analyses, from the simplest where all trial-level estimates were pooled (ignoring treatment comparison and index condition), through models where there was partial pooling between either index conditions or treatment comparisons, to the most complex model where the estimates were partially pooled across both treatment comparisons and index condition. The Bayesian models were fitted using the *brms* package [41] in R statistical software. Samples from the posterior distribution for each association was summarised as the mean and the 2.5th and 97.5th percentiles (credible intervals) of the posterior distribution. The proportion of the distribution less than 0 (probability of negative association, i.e., Bayesian *P*) was also reported.

(ii) *Association between comorbidity count and change in quality of life at trial follow-up*

We obtained estimates similarly for the association between comorbidity count and change in health-related quality of life from baseline to trial follow-up by fitting the same models but with final quality of life score as the outcome and a term for the baseline score in the trial-level linear regression models.

(iii) *Association between comorbidity count and the effect of treatment on change in quality of life at trial follow-up*

Lastly, we obtained estimates for comorbidity count–treatment interactions for change in health-related quality of life from baseline to trial follow-up by fitting the same model as (ii) with additional terms for treatment arm and an arm–comorbidity count interaction. We repeated this model in a sensitivity analysis after excluding trials that did not demonstrate a benefit in quality of life.

In order to allow other researchers to use the treatment–covariate interaction results to inform subsequent analyses (e.g., as an informative prior), we obtained samples from the posterior. We obtained samples for index conditions and treatment comparisons included in our model, as well as for a notional new index condition and new treatment comparison not included in our model. We summarised these samples as student t-distributions. As with the main analysis, these models were fit using the *brms* package (S1 Modelling description for additional details).

## Ethics approval and consent to participate

Ethical approval was obtained from the University of Glasgow, College of Medicine, Veterinary and Life Sciences ethics committee (200160070).

## Results

### Included trials

Of the 124 trials meeting our criteria [30,32], 56 provided 1 or more quality of life measures. Twenty-five trials (19,070 participants) provided EQ-5D only, 21 trials (8,595 participants) provided SF-36 only, and 10 trials provided both measures (5,756 participants) (Fig 1). One further trial that had collected EQ-5D and SF-36 was excluded as these results had been redacted by the study sponsors as part of the anonymisation process [42].

The characteristics of the 56 included trials were summarised by index condition (Table 1). The clinicaltrials.gov national clinical trial (NCT) identifiers of the included trials can be found in the Supporting information (Table A in S1 Additional figures tables). Index conditions were axial spondyloarthritis, chronic idiopathic urticaria, dementia, type 2 diabetes mellitus, pulmonary hypertension, inflammatory bowel disease, migraine, osteoporosis, Parkinson's disease, psoriasis, psoriatic arthropathy, pulmonary fibrosis, restless leg syndrome, rheumatoid arthritis, systemic lupus erythematosus (SLE), and thromboembolism. Not all arms were analysed from multi-arm trials; therefore, although included trials were limited to those with ≥300 participants, the number of analysed participants per trial ranged from 102 to 2,568 participants. Approximately 82.1% of trials had a placebo or usual care arm as the comparator. The follow-up ranged from 6 to 104 weeks. Mean follow-up was 34.5 weeks. Thirty-seven trials and 19 trials were analysed on the CSDR and YODA platforms, respectively. The mean age of trial participants ranged from 38 years in SLE trials to 73 years in dementia trials and the percentage of male participants ranged from 6% in SLE trials to 79% in pulmonary fibrosis trials.

As we had access to IPD, we were able to make the following assessments of trial robustness to possible biases: 94.6% were at least double blinded (double blind $n = 26$, triple blind $n = 8$, quadruple blind $n = 19$), there being only 3 open label trials. Participant completion ranged from 38.4% to 100%, with a mean (SD) completion of 81.1% (13.2%) (median 83.3% and IQR 71.4% to 90.6%). Given this is an IPD analysis there was no reporting bias.

### Comorbidity counts

As published previously, participants with comorbidities were present in trials for all index conditions [30]. Trials of selective immunosuppressants in SLE had the lowest proportion of participants with no comorbidities (6%) and the highest proportion of participants with ≥2 comorbidities (66%). Trials of tumour necrosis factor α (TNF-α) inhibitors in rheumatoid arthritis had the lowest proportion with ≥2 comorbidities (12%). Over half of participants in trials with SLE, pulmonary fibrosis, osteoporosis, pulmonary hypertension, and chronic idiopathic urticaria as the index condition, had 2 or more comorbidities.

### Comorbidity count and quality of life at baseline

The mean baseline EQ-5D index scores and SF-36-PCS and SF-36-MCS scores varied across conditions and intervention drug classes (Table 1). On average, participants in trials of TNF-α drugs in rheumatoid arthritis had the lowest mean baseline EQ-5D index value at 0.74 (0.08). The highest mean baseline EQ-5D index value was found in participants from trials in type 2 diabetes mellitus of glucagon-like peptide-1 receptor agonists and sodium glucose co-transporter 2 inhibitors 0.92 (0.09).

The lowest SF-36 component scores were seen in SLE trials of selective immunosuppressants with a mean SF-36-PCS 27.9 (2.5) and SF-36-MCS 20.68 (2.09). The highest SF-36 component scores, SF-36-PCS 66.42 (13.37) and SF-36-MCS 63.67 (13.89), were seen in psoriasis trials of interleukin inhibitors.

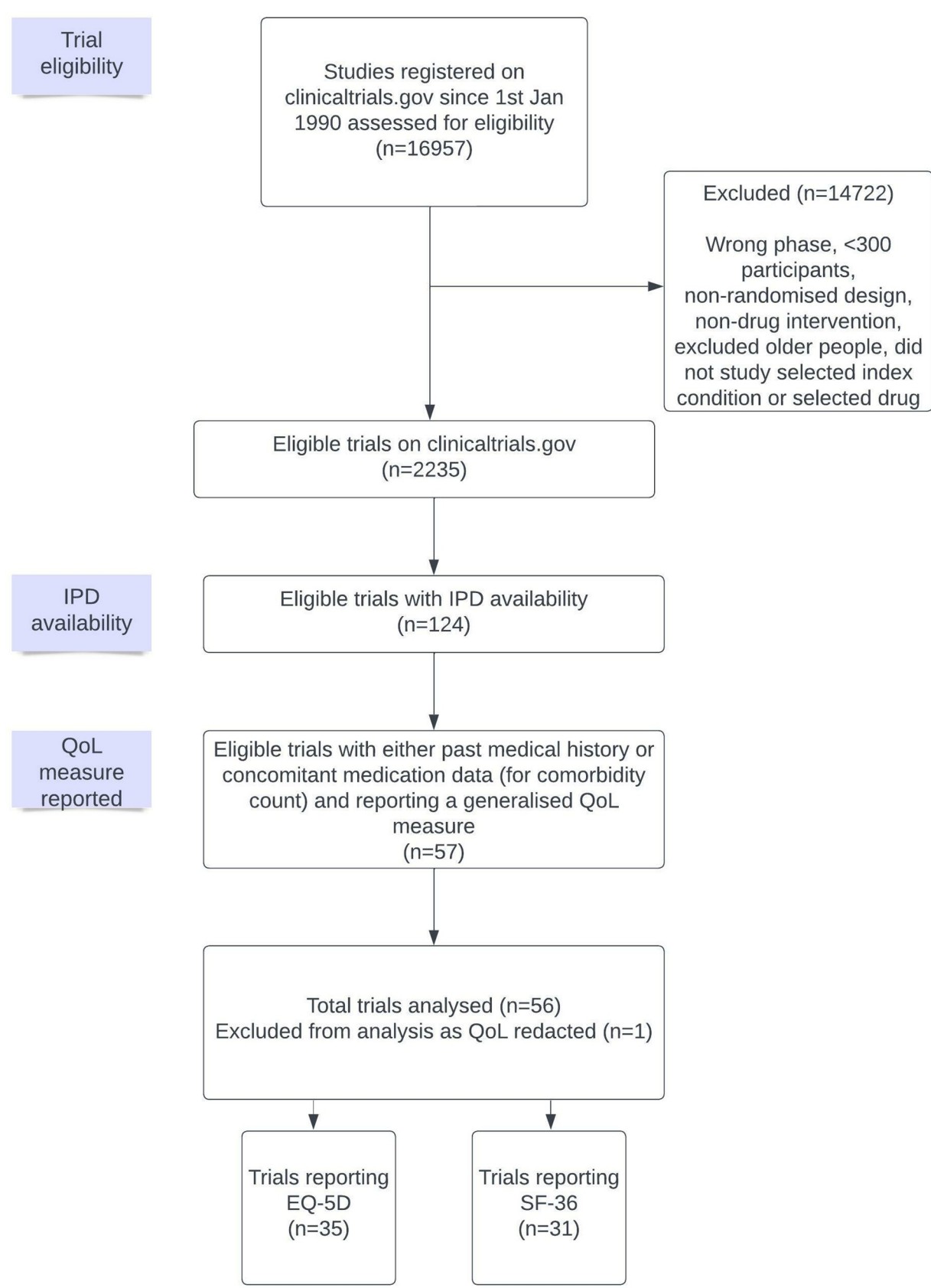

**Fig 1. PRISMA diagram for included trials in quality of life analyses.** Flow diagram of included trials with IPD and reporting QoL measures; EQ-5D or the 36-item short form survey (SF-36). IPD, individual participant data; QoL, quality of life.

Across all index conditions and treatment comparisons, there was a negative association between comorbidity count and quality of life at baseline, when adjusted for age and sex, for all 3 measures: EQ-5D effect estimate −0.04 standardised units (95% CI −0.05, −0.02; Bayesian $P > 0.999$) per additional comorbidity; SF-36-PCS effect estimate −0.13 standardised units (95% CI −0.2, −0.06; Bayesian $P > 0.999$) per additional comorbidity; SF-36-MCS effect

**Table 1. Participant characteristics by condition and drug class.**

| Trial index condition | QoL outcome | Drug treatment comparisons | n trials | Total participants | % Male | Mean age in years (SD) | Comorbidity count | | | Mean baseline quality of life |
|---|---|---|---|---|---|---|---|---|---|---|
| | | | | | | | 0 | 1 | ≥2 | EQ-5D index value (SD) or SF-36 PCS (SD); MCS (SD) |
| Axial spondyloarthritis | EQ-5D | L04AC | 1 | 102 | 75% | 42 (12) | 57% | 24% | 20% | 0.76 (0.08) |
| | SF-36 | L04AB, L04AC | 2 | 320 | 72% | 40 (12) | 49% | 32% | 19% | 43.18 (11.46); 47.62 (13.95) |
| Chronic idiopathic urticaria | EQ-5D | R03DX | 3 | 653 | 28% | 43 (14) | 19% | 27% | 55% | 0.89 (0.08) |
| Dementia | EQ-5D | A10BG | 3 | 2,265 | 41% | 73 (8) | 26% | 30% | 44% | 0.88 (0.09) |
| Diabetes mellitus, type 2 | EQ-5D | A10BJ, A10BK | 9 | 4,446 | 54% | 57 (10) | 31% | 30% | 40% | 0.92 (0.09) |
| | SF-36 | A10BK | 3 | 1,615 | 52% | 56 (9) | 26% | 25% | 49% | 55.45 (9.01); 50.46 (7.64) |
| Hypertension, pulmonary | EQ-5D | G04BE | 1 | 160 | 23% | 53 (15) | 22% | 22% | 55% | 0.83 (0.09) |
| | SF-36 | G04BE | 1 | 160 | 23% | 53 (15) | 22% | 22% | 55% | 44.98 (11.93); 54.87 (13.34) |
| Inflammatory bowel disease | EQ-5D | L04AB | 2 | 1,175 | 54% | 40 (13) | 43% | 32% | 24% | 0.85 (0.08) |
| | SF-36 | L04AA, L04AB, L04AC | 7 | 3,455 | 51% | 39 (13) | 42% | 33% | 25% | 49.67 (11.46); 44.12 (12.97) |
| Migraine | SF-36 | N03AX | 1 | 222 | 11% | 40 (11) | - | - | - | 63.48 (12.55); 59.9 (14.06) |
| Osteoporosis | EQ-5D | H05AA, M05BA | 4 | 4,377 | 45% | 68 (13) | 13% | 26% | 60% | 0.85 (0.11) |
| Parkinson's disease | EQ-5D | N04BC | 1 | 343 | 55% | 61 (10) | 44% | 32% | 24% | 0.78 (0.09) |
| Psoriasis | EQ-5D | L04AC | 3 | 1,117 | 68% | 45 (13) | 60% | 24% | 16% | 0.83 (0.15) |
| | SF-36 | L04AC | 2 | 960 | 68% | 46 (12) | 50% | 29% | 22% | 66.42 (13.37); 63.67 (13.89) |
| Psoriatic arthropathy | SF-36 | L04AB, L04AC | 3 | 597 | 55% | 47 (11) | 39% | 35% | 26% | 43.67 (13.51); 49.58 (15.67) |
| Pulmonary fibrosis | EQ-5D | L01XE | 2 | 1,062 | 79% | 67 (8) | 19% | 21% | 60% | 0.9 (0.1) |
| Restless legs syndrome | SF-36 | N04BC | 1 | 331 | 40% | 57 (12) | 24% | 34% | 42% | 63.55 (13.04); 60.79 (13.54) |
| Rheumatoid arthritis | EQ-5D | L04AB | 1 | 591 | 18% | 52 (12) | 61% | 27% | 12% | 0.74 (0.08) |
| | SF-36 | L04AB, L04AC | 9 | 4,608 | 20% | 51 (12) | 40% | 31% | 29% | 42.31 (12.35); 47.92 (14.31) |
| SLE | EQ-5D | L04AA | 2 | 1,112 | 6% | 38 (12) | 6% | 28% | 66% | 0.84 (0.1) |
| | SF-36 | L04AA | 2 | 1,126 | 6% | 38 (12) | 6% | 28% | 66% | 27.9 (2.5); 20.68 (2.09) |
| Thromboembolism | EQ-5D | B01AE | 3 | 6,450 | 59% | 55 (16) | 29% | 29% | 42% | 0.85 (0.11) |

For the 1 included migraine trial, % of participants per comorbidity count are redacted due to small numbers in the interest of anonymity. See Fig 2 for ATC drug class. ATC, anatomic therapeutic chemical; MCS, mental component score; QoL, quality of life; SD, standard deviation; SLE, systemic lupus erythematosus.

estimate −0.1 standardised units (95% CI −0.15, −0.04; Bayesian $P$ = 0.999) per additional comorbidity.

## Comorbidity count at baseline and subsequent change in quality of life

For all measures, relative to individuals without comorbidity, there was a decrease in quality of life over the course of the trial among participants with higher comorbidity counts. Table 2 shows these associations per one-unit increment in comorbidity count. The associations were similar in the simplest model where all trials were simply pooled and in the more complex models where trial was nested within index condition and/or drug class (Table 2). For EQ-5D, no model included the null (i.e., no association). For the SF-36 models, the results varied slightly by model. The 95% credible interval including the null for the model including drug class and index condition but did not include the null for the other models. However, even for the SF-36-PCS model, which had the widest 95% CI, the probability (Bayesian $P$ value) that comorbidity count was negatively associated with quality of life (i.e., there was a smaller improvement/greater fall in quality of life in those trials where quality of life improved/worsened, respectively) was 95.6%.

There was no evidence of departure from linearity for the association between comorbidity count and quality of life for any of the measures (Table B in S1 Additional figures tables). This means that for a participant with a comorbidity count of 1 compared to a participant with a comorbidity count of 0 (or indeed for any given one-unit increment in comorbidity count), the increase in quality of life during the trial was 0.02 standardised units lower for EQ-5D (difference −0.02; 95% CI −0.03 to −0.01; Bayesian $P$ value > 0.999), 0.05 units lower for SF-36-PCS, and 0.05 units lower for SF-36-MCS (Table 2). Back transforming these values to the original scales, this equates to 0.01 for the EQ-5D index, 0.48 for the SF-36-PCS, and 0.44 for

**Table 2. Change in quality of life measures from baseline to end of follow-up by comorbidity count.**

| Model complexity | Model adjustment | EQ-5D index value | SF-36-PCS | SF-36-MCS |
|---|---|---|---|---|
| | | 35 trials, 24,826 participants | 31 trials, 14,351 participants | 31 trials, 14,351 participants |
| All trials pooled | Unadjusted | −0.03 (−0.03 to −0.02); $P$ > 0.999 | −0.06 (−0.11 to 0); $P$ = 0.978 | −0.05 (−0.09 to 0); $P$ = 0.97 |
| | Age and sex adj. | −0.03 (−0.04 to −0.02); $P$ > 0.999 | −0.04 (−0.09 to 0.01); $P$ = 0.962 | −0.04 (−0.09 to 0.01); $P$ = 0.965 |
| Pooled by drug treatment comparisons | Unadjusted | −0.03 (−0.04 to −0.02); $P$ > 0.999 | −0.06 (−0.11 to 0); $P$ = 0.972 | −0.05 (−0.1 to 0); $P$ = 0.975 |
| | Age and sex adj. | −0.02 (−0.03 to −0.02); $P$ > 0.999 | −0.04 (−0.09 to 0.00); $P$ = 0.970 | −0.04 (−0.09 to 0.00); $P$ = 0.967 |
| Pooled by index condition | Unadjusted | −0.03 (−0.04 to −0.02); $P$ > 0.999 | −0.07 (−0.12 to −0.03); $P$ = 0.996 | −0.06 (−0.1 to −0.02); $P$ = 0.993 |
| | Age and sex adj. | −0.03 (−0.03 to −0.02); $P$ > 0.999 | −0.05 (−0.1 to −0.02); $P$ = 0.995 | −0.05 (−0.09 to −0.02); $P$ = 0.997 |
| Pooled by drug treatment comparisons and index condition | Unadjusted | −0.03 (−0.04 to −0.02); $P$ > 0.999 | −0.06 (−0.12 to 0); $P$ = 0.971 | −0.05 (−0.11 to 0.01); $P$ = 0.965 |
| | Age and sex adj. | −0.02 (−0.03 to −0.01); $P$ > 0.999 | −0.05 (−0.10 to 0.01); $P$ = 0.956 | −0.05 (−0.10 to 0.01); $P$ = 0.966 |

Meta-analyses for the association between comorbidity count and change in quality of life scores from baseline to trial follow up. Where both EQ-5D and SF-36 were reported in a single trial, trial number and participants are included in both totals. Standardised effect estimates, 95% credibility intervals and Bayesian $P$ values (probability of negative association).

MCS, mental component score; PCS, physical component score.

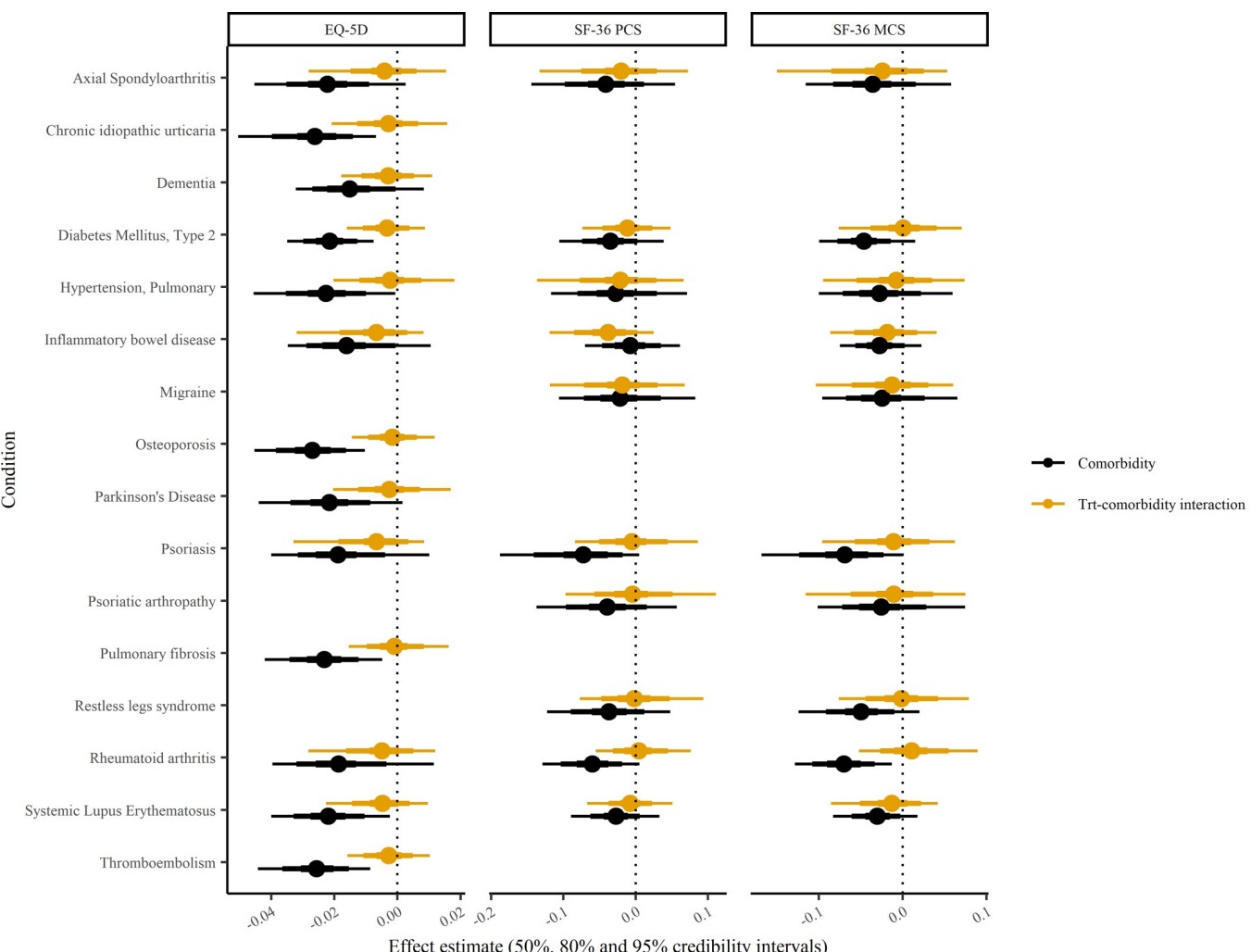

**Fig 2. EQ-5D, SF-36-PCS, and SF-36-MCS: change in quality of life and treatment interaction by condition.** The association of comorbidity and change in quality of life measures with treatment interaction (orange) and without (black). Standardised effect estimates with 50%, 80%, and 95% credibility intervals per trial index condition. EQ-5D, EuroQol-5 dimension; MCS, mental component score; PCS, physical component score; SF-36, the 36-item short form survey.

the SF-36-MCS. These associations between comorbidity count and change in quality of life are approximately similar in magnitude (per 1 additional comorbidity) to the treatment effects on change in quality of life for the 56 trials (Figure A in S1 Additional figures tables). Therefore, the presence of 2 or more comorbidities has a larger effect than treatment on quality of life. The associations were similar for different index conditions (Fig 2) and treatment comparisons (Fig 3).

## Comorbidity count at baseline and variation in the effect of treatment on quality of life during trial follow-up

There was no evidence of an interaction between comorbidity count and treatment effect on EQ-5D, SF-36-PCS, or SF-36-MCS. The probability that treatment was less effective in improving quality of life in those with a greater number of comorbidities (Bayesian *P* values) ranged from 0.556 to 0.749 (Table 3). The point estimates for these interactions were an order of magnitude smaller than those for comorbidity count and change in quality of life (see

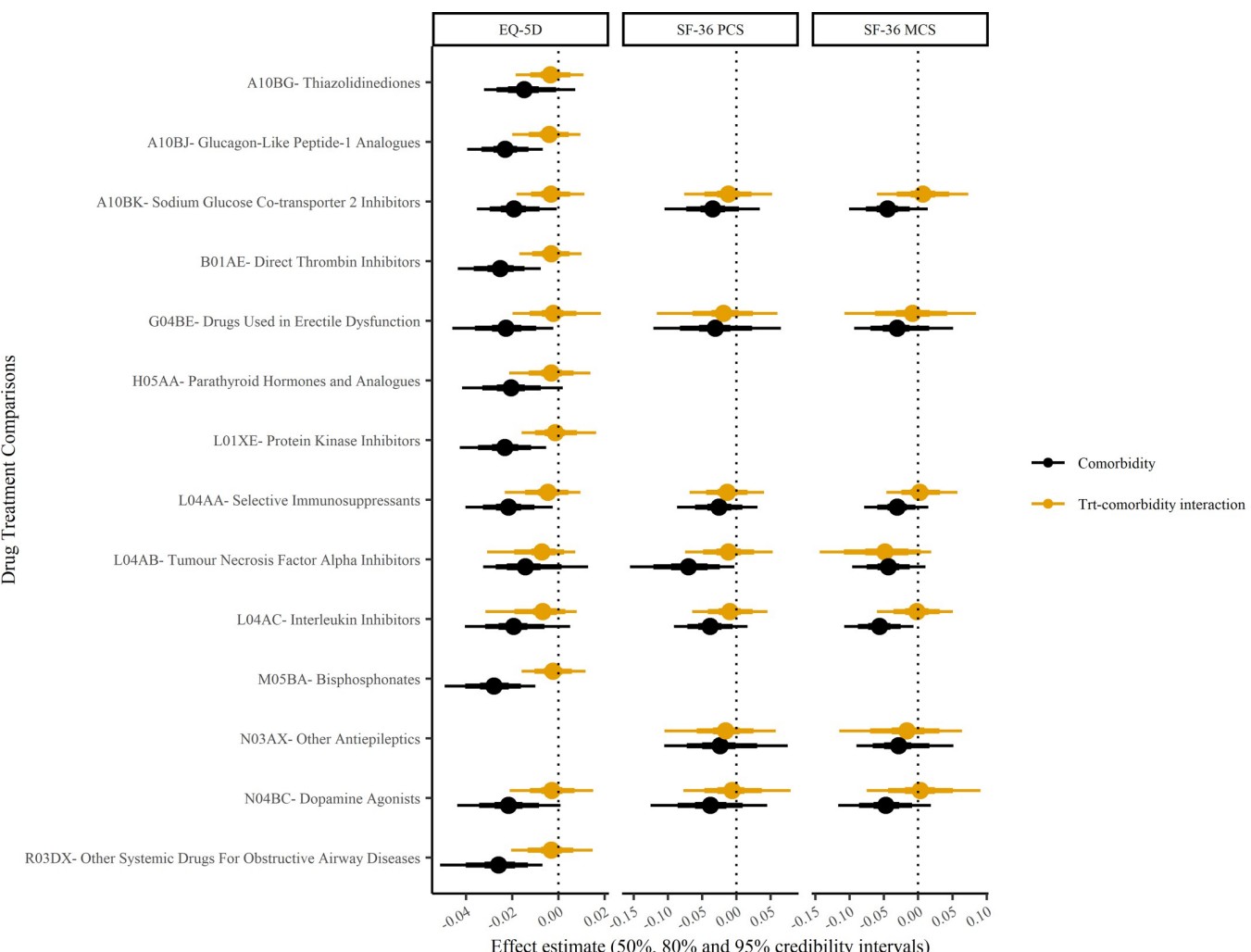

**Fig 3. EQ-5D, SF-36-PCS and SF-36-MCS: change in quality of life and treatment interaction by treatment comparison.** The association of comorbidity and change in quality of life measures with treatment interaction (orange) and without (black). Standardised effect estimates with 50%, 80%, and 95% credibility intervals per trial intervention ATC drug treatment comparisons. ATC, anatomic therapeutic chemical; EQ-5D, EuroQol-5 dimension; MCS, mental component score; PCS; physical component score; SF-36, the 36-item short form survey.

previous section). This was true across index conditions (Fig 2) and treatment comparisons (Fig 3), and the associations were null across all models from the simplest where all trials were pooled to the most complex where trial was nested within treatment comparison and index condition. Summarised posterior predictions (as student t-distributions) can be found in Table C in S1 Additional figures tables, for use by other researchers as informative priors for subsequent analyses. We also repeated the interaction analysis after excluding the trials (23 EQ-5D, 20 SF-36-PCS, and 23 SF-36-MCS) that did not demonstrate a benefit in quality of life; the findings were similarly null (Table D in S1 Additional figures tables).

## Discussion

In an IPD meta-analysis of 56 clinical trials across 16 index conditions and 14 treatment comparisons, we found that a higher comorbidity count at baseline was associated with poorer quality of life at baseline (as measured by EQ-5D, SF-36-PCS, and SF-36-MCS). The higher

**Table 3. Variation in the effect of treatment on quality of life by comorbidity count.**

| Model complexity | EQ-5D index value | SF-36-PCS | SF-36-MCS |
|---|---|---|---|
| | 35 trials, 24,826 participants | 31 trials, 14,351 participants | 31 trials, 14,351 participants |
| All trials pooled | −0.0023 (−0.0091 to 0.0047) $P = 0.747$ | −0.0042 (−0.0541 to 0.0409) $P = 0.574$ | −0.0105 (−0.0508 to 0.0234) $P = 0.749$ |
| Pooled by drug treatment comparisons | −0.0028 (−0.0117 to 0.0058) $P = 0.746$ | −0.0043 (−0.0517 to 0.0375) $P = 0.583$ | −0.0121 (−0.0536 to 0.0223) $P = 0.763$ |
| Pooled by index condition | −0.0029 (−0.0126 to 0.0056) $P = 0.743$ | −0.0027 (−0.0375 to 0.026) $P = 0.556$ | −0.0115 (−0.0557 to 0.0288) $P = 0.742$ |
| Pooled by drug treatment comparisons and index condition | −0.0035 (−0.0153 to 0.0065) $P = 0.746$ | −0.0092 (−0.0758 to 0.0476) $P = 0.631$ | −0.0111 (−0.0647 to 0.0416) $P = 0.70$ |

Meta-analyses for the interaction between comorbidity count and the intervention treatment effect on quality of life scores, where all models adjusted for age and sex. Where both EQ-5D and SF-36 were reported in a single trial, trial number and participants are included in both totals. Standardised effect estimates, 95% credibility intervals and Bayesian *P* values (probability of negative association).

MCS, mental component score; PCS, physical component score.

comorbidity count at baseline also predicted less improvement in EQ-5D over time. The findings were similar, but with wider 95% CIs which in some models just included the null, for SF-36-PCS and SF-36-MCS. However, baseline comorbidity count was not associated with differences in the estimated effect of treatment on quality of life for any of these measures.

Ours is the largest study, to our knowledge, to examine whether comorbidity predicts change in quality of life over time or in response to treatment in trial participants [24–29]. However, there are several limitations. First although comorbidity must, by definition increase with increases in multimorbidity, a simple count such as we used does not capture the full complexity of multimorbidity, including issues such as interactions between concordant and discordant conditions [43]. It is plausible that more nuanced measures of multimorbidity, feasible in trials designed to collect multimorbidity information prospectively, would lead to more nuanced results as to, for example, the effect of specific combinations of conditions on quality of life and heterogeneity of treatment effects. Moreover, while there was no evidence of any departure from linearity with the range of comorbidity counts we observed, this may not be true for higher comorbidity counts. Similarly, care should be taken when applying our findings to individuals with lower quality of life pretreatment, since the quality of life scores in the trial participants were generally fairly high at baseline. Secondly, the trials analysed were only those available via the CSDR and YODA trial repositories. Not all sponsors share data via these repositories, and among sponsors who do, not all trials are shared. Therefore, our dataset is not representative of all clinical trials, even for those index conditions and treatment comparisons included. As trials were identified from the US clinical trials register (clinicaltrials.gov) rather than a database of published papers (e.g., PubMed), there was no risk of publication bias. However, this does not mean that the trials were a random or complete sample of all registered trials. As we have noted previously [44], this set of trials were broadly similar to trials where IPD was not available with respect to the set indications, phases, number of participants, start dates, and exclusion criteria. However, inflammatory bowel disease and arthritis trials and trials of immunosuppressants were overrepresented, while phase 4 trials and those with especially large enrolment sizes were underrepresented.

Thirdly, while our definitions were prespecified, based on high-quality data, and have previously been used to demonstrate associations between comorbidity and both trial withdrawal and trial serious adverse events [5,31], the trials were not designed to measure comorbidity. It is possible that stronger associations may have been found if bespoke measures had been

available. Thirdly, few of the trial participants had more than 2 comorbidities. As such, care should be taken in extrapolating our findings to individuals with 3 or more conditions as well as to people with more severe comorbidities who are likely to be excluded from trials, since it is possible that these types of comorbidities do modify treatment responses. Finally, our analysis was designed to examine variation in treatment effects by comorbidity count not overall treatment effects. In our modelling, we shared information (partial pooling) across multiple index conditions and treatment comparisons, pooling more extensively than would have been appropriate had our focus been on specific treatment comparisons. Consequently, our analysis of comorbidity treatment interactions should not be used to make inferences about the overall efficacy of specific treatment comparisons, but rather about plausible variation in treatment effects by comorbidity count.

A number of previous studies have examined cross-sectional associations between multimorbidity (or the presence of multiple comorbidities and quality of life, consistently showing negative associations [8–22]). Two systematic reviews with meta-analyses demonstrated a negative association; one looked at studies in any setting (mean reduction in quality of life for each additional comorbid condition −1.55% (95% CI −2.97% to −0.13%) and −4.37% (95% CI −7.13% to −1.61%) for mental and physical quality of life, respectively) [21], and one examined studies in non-hospital settings (reduction in quality of life 3.8% to 4.1% per additional comorbid condition) [22]. Multimorbidity has also been found to be associated with lower quality of life in the USA Medical Expenditure panel survey [19], a population-based survey of cancer patients in the Netherlands [20], a cross-sectional study of 1,649 people in primary care in India [8], and among 3,256 people with spinal osteoarthritis identified from Korean national health survey data [14].

In contrast, we found only a small number of studies that examined whether comorbidity or multimorbidity at baseline predicted longitudinal change in quality of life. In a cohort study of 1,211 adults in Japan, multimorbidity at baseline predicted more rapid decline in SF-36 over 12 months [27]. Similarly, among 1,582 people undergoing total hip arthroplasty in Denmark comorbidity at baseline was associated with a smaller increase in quality of life (high comorbidity versus no comorbidity EQ-5D change 0.09; 95% CI 0.02 to 0.16) [28]. In a study comprising 351 individuals attending an Australian clinic for people with complex chronic diseases, comorbidity count predicted more rapid decline in SF-36 (−0.11 per comorbidity; −0.96 to 0.76) [29]. However, all 3 studies had substantial loss to follow-up. In a secondary analysis of a clinical trial including 379 participants with early rheumatoid arthritis, quality of life measures were regressed on multimorbidity (measured via the rheumatoid arthritis comorbidity index) using linear mixed models. SF-36-PCS was found to decrease more quickly among participants with multimorbidity but there was no association for SF-36-MCS [25]. Two studies reported multimorbidity at baseline and reported longitudinal measures of quality of life in people with head and neck cancer and prostate cancer, respectively, but neither presented effect estimates or 95% confidence intervals making it difficult to judge whether this was due to the low sample sizes [24,26]. To this literature, we add the observation that in 33,421 participants in 56 trials across 16 index conditions, baseline multimorbidity is associated with lower quality of life at baseline and—at least for the EQ-5D-index—less improvement in quality of life associated with trial participation.

While we did not undertake a formal systematic review, in a wide-ranging search including a number of terms for comorbidity, multimorbidity, and quality of life (Table E in S1 Additional figures tables), we found only 1 study that reported findings on variation of treatment effects on quality of life by comorbidity or multimorbidity. In a convenience sample of 3 trials (1 asthma, 1 heartburn, and 1 gastric ulcer disease) [45], the reported associations differed across the included trials, measures of quality of life and measures of multimorbidity. To this

sparse literature, we add findings, from a large number of participants, that there was no evidence of heterogeneity of treatment effect by multimorbidity for EQ-5D, SF-36-PCS, or SF-36-MCS.

The observation that treatment effects on quality of life did not differ by multimorbidity (measured via a comorbidity count) at baseline has implications for the interpretation of clinical trials. Unless there are strong a priori reasons to believe that an effect of a treatment on quality of life will be differential by comorbidity (e.g., a trial of a diabetes drug given by injection in a device that might be more difficult for people with arthritis to use may lead to differential efficacy in those people), reports of heterogeneity from individual trials and small meta-analyses should most likely be considered chance findings. Similarly, treatments that involve complex regimes may also differentially impact quality of life according to the participant's comorbidity count. For example, individuals with more comorbidities (and hence greater polypharmacy) may find additional treatments either more or less burdensome than individuals with fewer comorbidities. Our findings imply that treatment efficacy estimates for EQ-5D, SF-36-PCS, and SF-36-MCS from clinical trials are also likely to be applicable to settings with (at least modestly) higher levels of comorbidity, at least for the kinds of conditions and drug classes covered in our analysis.

Based on our models, for each outcome, we produced posterior predictions for comorbidity–treatment interactions for an unobserved drug class and index condition (i.e., one not included in our models). These distributions represent an estimate of the likely variation in treatment effect according to comorbidity count before seeing the trial data for any given specific treatment comparison. We had originally intended to repeat the modelling after log-transforming quality of life measures. However, since there were no interactions on the linear scale (which is standard for these measures), we did not examine the effect of changing scale. As such, these distributions can be used as informative priors for subsequent meta-analyses examining treatment effects in people with comorbidity, improving the reliability and precision of the resultant estimates. The prior distributions we produced can also be used to inform choices in probabilistic health economic models that are used to apply trial findings to "real-world" populations, such as in Health Technology Assessments [46]. For example, if a trial was conducted among participants with fewer comorbidities than patients in the target population, our findings can be used to model the likely effect of such treatments on quality of life in real-world settings. We observed a decline in quality of life in participants with comorbidities, relative to other participants. This finding provides new evidence to suggest that the association between comorbidity and quality of life is causal. Most previous studies reporting associations between multimorbidity or comorbidity and quality of life have been cross-sectional [8–22], and the few longitudinal studies have been limited by small sizes [24–26], limited numbers of baseline conditions, or substantial loss to follow-up [27–29]. Our findings therefore strengthen the evidence for causation between comorbidity and baseline and subsequent change in quality of life.

There are a number of possible mechanisms for the relative decline in quality of life among people with multiple comorbidities. First, the decline may be a direct result of the underlying conditions (alone or in combination); symptoms such as pain, impaired sleep, limited mobility, and impaired function may increasingly adversely impact quality of life over time. Alternatively, the (non-treatment related) benefits of trial participation such as improved access to clinical care may be different in people with and without comorbidity. Similarly, attending visits, undergoing procedures, and following treatment regimens may also impose a greater treatment burden among people with comorbidity. This concept of trial participation burden is analogous to the treatment burden (visits, drug regimens, etc.) described in routine clinical practice [47], which is known to be more challenging for people with multimorbidity.

In these clinical trials, higher comorbidity count is associated with lower quality of life at baseline and predicts subsequent relative decline in EQ-5D (and most likely SF-36) over time. However, the effect of treatment on quality of life does not differ by comorbidity count at baseline. Trial-derived estimates for treatment effects for quality of life are likely to be applicable to people with moderate numbers of comorbidities.

## Supporting information

**S1 PRISMA checklist. PRISMA checklist.** This file is a PRISMA checklist for the manuscript.
(DOCX)

**S1 Modelling description. Detailed description of modelling. This file includes a further detailed description of the statistical modelling performed.**
(DOCX)

**S1 Additional figures tables. Supplementary figures and tables. This file contains supplementary figures and tables.**
(DOCX)

## Acknowledgments

This study, carried out under YODA Project # 2017–1746, used data obtained from the Yale University Open Data Access Project, which has an agreement with JANSSEN RESEARCH & DEVELOPMENT, L.L.C. The interpretation and reporting of research using this data are solely the responsibility of the authors and does not necessarily represent the official views of the Yale University Open Data Access Project or JANSSEN RESEARCH & DEVELOPMENT, L.L.C. This study was also carried out under ClinicalStudyDataRequest.com project number 1732, used data from the ClinicalStudyDataRequest.com repository, who provided data from Boehringer-Ingelheim, GSK, Lilly, Roche, Takeda, and Sanofi. The interpretation and reporting of research using these data are solely the responsibility of the authors and does not necessarily represent the official views of ClinicalStudyDataRequest.com or Boehringer-Ingelheim, GSK, Lilly, Roche, Takeda, or Sanofi.

## Author Contributions

**Conceptualization:** Peter Hanlon, Anoop S. V. Shah, Emma McIntosh, Jim Lewsey, Sarah H. Wild, Bruce Guthrie, David M. Kent, Sofia Dias, Nicky J. Welton, David A. McAllister.

**Data curation:** Elaine W. Butterly, Laurie J. Hannigan, David A. McAllister.

**Formal analysis:** Elaine W. Butterly, David A. McAllister.

**Funding acquisition:** David A. McAllister.

**Investigation:** David A. McAllister.

**Methodology:** Emma McIntosh, Sarah H. Wild, Frances S. Mair, Sofia Dias, Nicky J. Welton, David A. McAllister.

**Project administration:** Elaine W. Butterly, David A. McAllister.

**Resources:** David A. McAllister.

**Software:** David A. McAllister.

**Supervision:** David A. McAllister.

**Validation:** David A. McAllister.

**Visualization:** David A. McAllister.

**Writing – original draft:** Elaine W. Butterly.

**Writing – review & editing:** Elaine W. Butterly, Peter Hanlon, Anoop S. V. Shah, Laurie J. Hannigan, Emma McIntosh, Jim Lewsey, Sarah H. Wild, Bruce Guthrie, Frances S. Mair, David M. Kent, Sofia Dias, Nicky J. Welton, David A. McAllister.

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
