## [Editor Report · Decision Letter 0]

29 Jun 2022

Dear Dr McAllister, 

Thank you for submitting your manuscript entitled "Comorbidity and health related quality of life in randomised clinical trials: an individual participant data meta-analysis" for consideration at PLOS Medicine.

Your manuscript has now been evaluated by the PLOS Medicine editorial staff as well as by an academic editor with relevant expertise and I am writing to let you know that we would like to send your submission out for external peer review.

Please re-submit your manuscript within two working days, i.e. by Jul 01 2022 11:59PM.

Kind regards,

Dr Philippa Claire Dodd, MBBS MRCP PhD

Senior Editor

PLOS Medicine

---

## [Decision Letter · Decision Letter 1]

26 Sep 2022

Dear Dr. McAllister,

Thank you very much for submitting your manuscript "Comorbidity and health related quality of life in randomised clinical trials: an individual participant data meta-analysis" (PMEDICINE-D-22-02156R1) for consideration at PLOS Medicine. 

[LINK]

In light of these reviews, I am afraid that we will not be able to accept the manuscript for publication in the journal in its current form, but we would like to consider a revised version that addresses the reviewers' and editors' comments. Obviously we cannot make any decision about publication until we have seen the revised manuscript and your response, and we plan to seek re-review by one or more of the reviewers. 

We expect to receive your revised manuscript by Oct 17 2022 11:59PM. Please email us (plosmedicine@plos.org) if you have any questions or concerns.

We look forward to receiving your revised manuscript. 

Sincerely,

Philippa Dodd, MBBS MRCP PhD

PLOS Medicine

pdodd@plos.org

plosmedicine.org

GENERAL

As per our email correspondence, please include the code used to determine mortality scores. See also reviewer comments below.

Please report your SR/MA according to the PRISMA guidelines provided at the EQUATOR site. http://www.equator-network.org/reporting-guidelines/prisma/

Please provide the completed PRISMA checklist. When completing the checklist, please use section and paragraph numbers, rather than page numbers. 

Please add the following statement, or similar, to the Methods: "This study is reported as per the Preferred Reporting Items for Systematic Reviews and Meta-Analyses (PRISMA) guideline (S1 Checklist)."

Your search end dates are not clear (until one reaches the suppl files) please include in the abstract & main manuscript (see below)

Please update your search to the present time (within the last 6 months, if not already).

Please embed abbreviations within the main manuscript text at the time the abbreviation is first referred to and remove from the end of the manuscript. Please ensure inclusion of abbreviations where relevant in all tables/figures 

Please remove consent/data availability statement/competing interests statement from the end of the manuscript and include in the manuscript submission form instead

ABSTRACT

Please report your abstract according to PRISMA for abstracts, following the PLOS Medicine abstract structure (Background, Methods and Findings, Conclusions) http://www.plosmedicine.org/article/info:doi/10.1371/journal.pmed.1001419 .

Please combine the Methods and Findings sections into one section, “Methods and findings”.

Abstract Methods and Findings: 

Please ensure that all numbers presented in the abstract are present and identical to numbers presented in the main manuscript text. 

Please provide the dates of search, data sources, types of study designs included (specify RCT results line 1)

Please quantify the main results p values, as well as with 95% CIs. 

Please include the important dependent variables that are adjusted for in the analyses. 

In the last sentence of the Abstract Methods and Findings section, please describe the main limitation(s) of the study's methodology.

AUTHOR SUMMARY

INTRODUCTION

Line 69: Please replace “background” with “Introduction”

METHODS and RESULTS

Please quantify the main results p values, as well as with 95% CIs, where relevant

Where p-values are reported, please also provide the statistical tests used to determine them 

Please report the end date of your search - line 108: “…from January 1990 onwards…”

Please evaluate study quality and risk of bias.

Please evaluate evidence of publication bias.

Please clearly report how you searched for and selected published studies including the search platforms

Please ensure that your ethics statement is included in the manuscript methods section, as stated in the submission form, and remove from the end of the manuscript

Line 106: “…(Prospero CRD42018048202).(30) The full selection process and analysis plan has been documented previously. (31)” please include the analysis plan with the manuscript, see also reviewer comments below

DISCUSSION

Thank you for organizing the discussion largely as follows: a short, clear summary of the article's findings; what the study adds to existing research and where and why the results may differ from previous research; strengths and limitations of the study; implications and next steps for research, clinical practice, and/or public policy. Please remove the sub-headings from the discussion and place the conclusion as the final paragraph (removing the heading “conclusion”). 

FIGURES

Thank you for including the figure captions and the figures. Please also place these within the manuscript results section

I agree with the reviewer comments below that the PRISMA flowchart would be a helpful addition to the main manuscript.

Please consider avoiding the use of red (and green) to make the figures accessible to those with colour blindness

TABLES

Table 2 – please also provide unadjusted analyses

REFERENCES

Please select the PLOS Medicine reference style in your citation manager. In-text reference call outs Citations should be in square brackets, and preceding punctuation, note the absence of spaces within the square brackets, “…symptomatic [2,8].” 

In the bibliography please ensure no more than 6 authors are listed before et al where more than n6 authors contribute

Journal name abbreviations should be those found in the National Center for Biotechnology Information (NCBI) databases. 

Comments from the reviewers:

Reviewer #1: Thanks for the opportunity to review your manuscript. My role is as a statistical reviewer, so my review concentrates on the study design, data, and analysis that are presented. I have put general questions first, followed by queries relevant to a specific section of the manuscript (with a page/line reference).

The manuscript presents and IPD meta-analysis, of data drawn from pharma sponsored RCTs that collected both a HRQOL measure and a comorbidity count. The data is used to assess two questions, 1) whether higher comorbidity count is associated with greater declines in HRQOL over time (from baseline of study to a follow-up point), and 2) whether there is an interaction between treatment effects (on HRQOL) and baseline comorbidity (i.e. a subgroup analysis for HRQOL outcome). A two-stage approach (analysis on each trial, then synthesis of the estimates from each trial) was used. This approach was used over a one-stage (multilevel model) approach because of the availability of trial data stored across different platforms. The relevant coefficients and SEs were estimated with linear regression, then in the second synthesis stage, these were pooled using a Bayesian LM. The 'ANCOVA' approach to estimating change from baseline is used for the interaction between treatment effects and baseline comorbidity. Clearly a lot of work went into doing the first stage of the analysis. 

The study is registered on PROSPERO, and the approach detailed in this manuscript matches the registration. One thing I was curious about was whether you kept any data about how comorbidities were recorded across all of the trials. My own experience suggests a high degree of heterogeneity when you go from one therapeutic area to another (e.g. renal trialists seem to try to capture everything), and wondered if this might account for some of the differences in comorbidity counts by type of disease.

Is it possible to see a copy of the code that derives the comorbidity count? Usually during the review process names are concealed so granting a view request might not work, but is it too unwieldy to attach the syntax.

I would consider some adjustment to the title if there is space, in particular to reflect that the study is focused on drug trials of chronic disease patients.

With only a limited number of data sets available for IPD, do you consider the results to be generalizable to multi-morbidity in all drug RCTs? 

P6, L108. One important detail I would add here from the PROSPERO protocol was the disease selection criteria i.e. that ID, cancer etc. are excluded

P9, L172. To clarify, this means that age was rescaled to be in units of SD?

P10, L174. Were any checks of residuals from this stage done? (I can understand that this would be a pain with so many trials to check). What criteria was used to decide that the polynomial (squared) term of # comorbidities was consistent with linearity? A second order polynomial is a decent way to account for non-linearity but it won't always work in my experience.

P12, L252. I think this is an accurate statement about the main results, one thing I struggled with was translating what the coefficient (i.e. change in HRQOL with one extra comorbidity) translates to. I didn't have a sense of whether this was a very dramatic change or a minor one. I am not sure if this can be changed easily - and might just reflect that I don't usually work with HRQOL outcomes more than any problem with the manuscript. 

P13, L280. This is the posterior probability of a negative interaction beta coefficient? 

Supp Appendix. I would describe the prior for the overall intercept to be 'weakly informative' (SD=10), but it looks like the trial/cond/comp priors are more strongly informative (SD=1). What was the rationale of this choice (or have I misunderstood this part?)? 

Reviewer #2: Thanks for the opportunity to review this IPD that aims to improve understanding about the impact of multimorbidity in HRQoL (and whether multimorbidity affects treatment effects on HRQoL).

It is a policy relevant piece of research that has potential to contribute to understanding about the causal relationship between multimorbidity and QoL. 

I don't have any major reservations about the core methods deployed and these are mainly well described.

However I was struck by the conceptual inconsistency that perhaps undermines the thrust of the analyses and the arguments off the back of the findings.

1. In the Introduction the study is contextualised in relation to multimorbidity, characterised as two or more conditions. But in the methods the authors pivot to talk about comorbidity and this seems to represent significant conceptual slippage that is not adequately addressed. Their approach is fundamentally about co-morbidity where there is an index condition with an intention to then add up numbers of comorbidities, which essentially assumes comorbidity is additive. But this approach (as discussed by Ng et al doi: 10.1093/ije/dyy134 [and others]) does not account for multimorbidity that occurs by chance and is reliant on the range of illnesses in the dataset. It does not take into account the way certain illness might cluster and have syntergistic affects on health outcomes and as such I wonder if the authors can address this and refer to literature on multimorbidity clusters, to more adequately defend their approach. See for example: Sinnige et al. doi:10.1371/journal.pone.0079641 and Busija https://doi.org/10.1007/s10654-019-00568-5; Prados-Torros http://dx.doi.org/10.1016/j.jclinepi.2013.09.021

2. The number of index conditions (21) seems quite small compared to previous epidemiological studies of multimorbidity e.g. Barnett et al. https://www.thelancet.com/journals/lancet/article/PIIS0140-6736(12)60240-2/fulltext

3. Excluding trials with populations younger than 60 risks excluding populations with multimorbidity from socio-economically disadvantaged backgrounds where multimorbidity is known to affect people 10 years younger (see for example https://www.thelancet.com/journals/lanhl/article/PIIS2666-7568(21)00146-X/fulltext)

4. Additionally, the list of index conditions excludes mental health problems, both common and serious (if we exclude dementia as not belonging to that family of clinical health problems). This seems to be a major omission given the prevalence of mental and physical multimorbidity and this should be explained and made clear in the title or at least the main description of the study.

Reviewer #3: Thank you for asking me to review this paper that read with considerable interest. I think there is a very important message here that has the potential to impact on important aspects of trial design and interpretation. That is that factors that predict outcome do not necessarily predict response to treatment. This means we have some reassurance that, at least in the context of the trials studied here, that effects observed in people with fewer co-morbidities are likely to be generalisable and can be applied clinically to those with more co-morbidities.

 I look forward to the team repeating these analyses, subject to data availability, for other baseline factors measured in clinical trials such as sex/gender, ethnicity, age, specific co-morbidities (e.g. depression). For the record I would still have liked the paper even if the results were different or inconclusive.

I have reviewed this paper from the perspective of a clinical trialist with experience of IPD meta-analysis for sub-group effects. Whilst I think a Bayesian approach is appropriate for work I am not competent to comment on the actual methods used. This would require specialist statistical review. 

I have some, largely minor comments

1. My understanding is that the Euroqol group do not consider EQ-5D to be an abbreviation and that this is the full name of the measure and so no need to define this as Euroqol-5 dimension.

2. EQ-5D-3L, EQ-5D-3L, and SF-36 are usually hyphenated in this style

3. This is one of several papers published by this group on this dataset. I think it would be helpful to briefly document these, and their headline conclusions in the background to make it clear where this paper sits in the overall body of the work

4. I think there is too much extraneous data in the abstract. Detailing the baseline relationship between co-morbidity & EQ-5D / SF-36 distracts from the main points of interest - that people who are less well do less well overall but that they have just as much to gain from effective treatments. It is the absence of any interaction between co-morbidity and that treatment efficacy that is the golden nugget from this paper. This could be drawn out more in the abstract. Also, for the more general reader I think some explanation is needed of 'standardised units'

5. We are directed to the authors previous paper documenting that co-morbidity in clinical trials is less common than in the general population as a source for the full study selection process and the analysis plan. This paper tells us more about the selection process but provides no information on the statistical analysis plan for this new paper. Further, the entry in Prospero is too limited to indicate the nature of the analysis plan. Here it states;

'For each trial, for each outcome, estimates of covariate treatment-interactions will be obtained. The resultant trial-level results will be combined in Bayesian hierarchical generalized linear model to estimate interactions at the level of drug-classes and wider groupings of related drug classes.'

This indicates that original focus was on effects within drug classes rather than the overall effects reported here. I think the full, a priori, statistical analysis plan should be available with this paper.

6. In lines 142 to 145 the authors allude to mapping between EQ-5D-5L and EQ-5D-3L and refer here to a standard look-up table. I assume here that hey are using the Van Hout algorithm. Thus I assume they are standardising to a UK value set for EQ-5D-3L. If so this should be clear. There should be a link to where this can be found directly rather than the generic Euroqol website. There is a link to this in the EQ-5D-5L manual - but the link is broken and so I was unable to quickly find the relevant details.

7. For this current paper this sentence on lines 155 - 156 is probably superfluous 'SF36 can be mapped to preference based measures such as EQ5D for use in economic evaluations.'

8. In table 1 do we really mean age down to two places of decimals? Might whole numbers be easier on the eye and just as informative? Similarly for percentage of males whole percentages are likely to just as informative as presenting to one decimal place. If authors disagree on this point they need to edit the 28% figure from R03DX study

9. In the review copy figures 1&2 were rather blurred. A clearer version will be needed for publication

10. I think the PRISMA flow chart in the supplementary files might be better included in the main paper

[LINK]

---

## [Decision Letter · Decision Letter 2]

14 Nov 2022

Dear Dr. McAllister,

Thank you very much for re-submitting your manuscript "Comorbidity and health related quality of life in people with a chronic medical disease in randomised clinical trials: an individual participant data meta-analysis" (PMEDICINE-D-22-02156R2) for review by PLOS Medicine.

I have discussed the paper with my colleagues and the academic editor and it was also seen again by 3 reviewers. I am pleased to say that provided the remaining editorial and production issues are dealt with we are planning to accept the paper for publication in the journal.

[LINK]

We look forward to receiving the revised manuscript by Nov 21 2022 11:59PM.   

Sincerely,

Pippa

Philippa Dodd, MBBS MRCP PhD

PLOS Medicine

plosmedicine.org

Requests from Editors:

Thank you for your considered and detailed responses to previous reviewer and editor comments (including those that may have appeared irrelevant) which are vital to maximizing the transparency of data. Please see below for further minor comments and suggested revisions.

COMMENTS FROM THE ACADEMIC EDITOR

A couple points in the article that were not entirely clear to me were: 

1) how the evidence that they have in the article/appendix about linear relationship between predictor and outcome is consistent with the conclusion that non-linear relationship is not present (as opposed to it is unclear whether the relationship is linear or not) -- the linear assumption may be important if there are some people with outlier # of comorbidities which can have a lot of leverage in a linear model's slope estimates; also trial exclusions may systematically take out some types of comorbidities which could be more influential and hence generalization from trial efficacy and efficacy modification by comorbidity to population effectiveness and effect modification may be more challenging

2) the baseline QoL in the trials is generally relatively high (almost all 0.80 or above for EQ4D QoL weight) so it is probably important to not overly confidently extrapolate too far out of sample (to patients with baseline QoL that is substantially below) in terms of the impact of comorbidities on change in QoL (e.g., just because comorbidities for relatively healthy people [high QoL at baseline] may not have a very strong impact on the size of a treatment effect does not necessarily imply that this will be the case for people with much lower QoL at baseline [whose QoL is substantially outside the range of the study populations in the trials considered])

GENERAL

Please address all reviewer and editor comments in full

Please check the in-text reference call-outs throughout the manuscript there is occasionally a space missing between the text and the opening parentheses e.g. line 103 “patients[1]”, line 112 “reported[8–22]” and so on. Please ensure that punctuation follows closing parentheses e.g. line 282 “…conditions. [30] Trials….” should read "…conditions [30]. Trials….” Please check and amend throughout.

TITLE 

suggest “conditions” in place of “diseases” or something similar

ABSTRACT

Thank you for updating the abstract and reporting according to PRISMA guidance. We note the comments from reviewer 3 regarding inclusion of information in the abstract and acknowledge that PRISMA reporting necessitates this for purposes of transparent data reporting. In context of the reporting guidance all findings should be reported in the abstract. 

To me, not dissimilar to reviewer 3, the finding of an absence of interaction between co-morbidity and treatment efficacy is more interesting and more novel than other findings and I suspect may be to others reading your manuscript also. I agree with reviewer 3 that with respect to this particular outcome, as written, the importance of this point is somewhat lost, particularly on the general reader. I would suggest re-wording/elaborating the final sentences of the methods and findings section (lines 57-59) as well as the abstract conclusion such that the meaning/implications of the treatment efficacy data are clearer especially to the more general reader – i.e. what does the negative association mean in real terms at the patient and/or policy level? (see also author summary below)

Line 59: “EQ5D or SF36” – should hyphenate here also, please check throughout and amend where necessary.

AUTHOR SUMMARY

Line 74: perhaps “physical and mental health” instead of “mental function and wellbeing”, or something similar

Line 78: beginning “Multimorbidity…” suggest “Multimorbidity, the presence of two or more conditions, makes diagnosis and treatment more complex and is associated with worse quality of life in some settings.” The first sentence, as written, is perhaps redundant?

Line 80: beginning “Moreover, people with multimorbidity…” suggest beginning as a separate bulleted point as “People with multimorbidity are….” 

What do these findings mean? 

In general, I think you undersell your study, perhaps in an attempt not to oversell it (which is always important). I would suggest you revise this section with the below comments in mind:

Line 96: “…which helps inform clinical decision making” – how? Please justify the statement

Line 93: “a higher comorbidity count did not change the effect of treatment on quality of life.” and line 98: “…suggest that the effect of treatments on quality of life scores…” It may help to explicitly/simply state what the effect of treatment is. Throughout this is somewhat left to assumption which may not be helpful to the general reader.

Line 97: “These findings also provide some reassurance for clinicians and guideline developers…” – again how? I would revise this point in line with the above, what is the effect of treatment on QoL scores? I think this could be structured/re-worded to pack a bigger punch – which I think it deserves – it’s important not to overstate findings but I think you’re somewhat understating/underselling the importance of your study outcomes here.

Line 100: “…and for the types of treatments and conditions included in this analysis.” Rather paradoxically the statement is both broad and vague. I appreciate it would be impossible to list these but is there a better way to umbrella the included conditions (or refer to categories of those excluded)? Perhaps not given the nature of the study, in which case would it be better removed altogether? I'll leave it to your discretion, but I would have a think about how this is written such that you include as much necessary information as possible.

METHODS and RESULTS

Line 278: “…mean (sd) completion” should this read (SD) if representing ?standard deviation

FIGURE 1

Thank you for including this. Line 261: please hyphenate QoL surveys EQ-5D and SF-36 in the caption

FIGURE 2

Thank you for kindly altering the colour schemes of your figures to improve accessibility to the reader with colour blindness. Your caption(s) still refer to your previous colour scheme “...(blue) and without (red)…”. Please amend according to the new scheme and throughout, including supplementary files, where relevant.

DISCUSSION

The discussion (and preceding results section) achieves much better clarity regarding the implications of your study outcomes than the abstract/author summary. I would encourage you to cross reference any specific changes you make to the abstract/author summary with these sections to ensure consistency of language and terms, thus interpretation, throughout the manuscript.

Line 369: “Ours is the largest study to examine whether…” please temper assertations of primacy – suggest “to our knowledge” or something similar

LANGUAGE

Please ensure the term disease is replaced with condition or something similar in the title and please check throughout for the same

SOCIAL MEDIA

Please include any twitter handles for your institution/funding body etc such that we can help to extend the reach of your research

Comments from Reviewers:

Reviewer #1: Thanks for the revised manuscript and responses to my review. Compiling the information into the GitHub repository made completing the re-review much easier. 

The sensitivity check with penalised splines is consistent with the polynomial terms, I think this looks fine to me. 

I don't think there are any changes needed with the coefficient (my question about interpretability of the beta coefficient), I think you are right that there aren't any benchmarks because it's almost always normally an input in a CE analysis (which has a more obvious interpretation). 

The explanation about the priors was helpful thank you - with the additional information available via the appendix I don't think there's any changes that need to be made here.

I happily recommend the manuscript be accepted.

Reviewer #2: Thank you for further reflecting on how to position this paper in the context of research on multimorbidity and comorbidity. I think the additional contributions clarifies the approach taken to conceptualising multimorbidity and justifying the selection of comorbidities.

Reviewer #3: I was pleased to see this revised article.

I must first apply some peer review to the editorial requirements with regard to how systematic reviews should be presented. To simplistically apply quality standards used for a conventional review to an IPD meta-analysis is simply wrong. This is an analysis on data are available in an IPD resources. It is simply not feasible to include trials published within six months. There is quite clearly no need to update the search to being within the previous six months. Further I suggest that conventional risk of bias assessment is also inappropriate. The authors are not here trying to suggest any main effects of an intervention. Rather they are looking at within trials data to help us better understand effects of co-morbidity. Above a very low bar for quality, that is met by having data of suitable quality to be included, and evidence of robust randomisation all these data will be of good quality. Essentially the data are what the data are

I remain very surprised that the authors think it a novel finding that people with multi-morbidity are less well, and have poorer outcomes is a novel finding. There is a vast amount of observational data, and indeed clinical experience, on this matter. I would also suggest that the highly selective nature of recruitment to the included trials means that these data are far inferior to the existing observational data on this point. Whilst these data need reporting the really novel finding from this work is that multi-morbidity does not moderate outcome, within this dataset. Thus I remain surprised that the authors report effect sizes for their first two analyses in the abstract when not including the (non) effect size for treatment moderation.

[LINK]

---

## [Editor Report · Decision Letter 3]

9 Dec 2022

Dear Dr McAllister, 

On behalf of my colleagues and the Academic Editor, Professor Jeremy Goldhaber-Fiebert, I am pleased to inform you that we have agreed to publish your manuscript "Comorbidity and health related quality of life in people with a chronic medical condition in randomised clinical trials: an individual participant data meta-analysis" (PMEDICINE-D-22-02156R3) in PLOS Medicine.

PRESS

Best wishes,

Pippa 

Philippa Dodd, MBBS MRCP PhD 

PLOS Medicine